# Developmental Screening Tools Used with First Nations Populations: A Systematic Review

**DOI:** 10.3390/ijerph192315627

**Published:** 2022-11-24

**Authors:** Sara Cibralic, Patrick Hawker, Feroza Khan, Antonio Mendoza Diaz, Susan Woolfenden, Elisabeth Murphy, April Deering, Clare Schnelle, Sharnee Townsend, Kerrie Doyle, Valsamma Eapen

**Affiliations:** 1Ingham Institute, Liverpool, NSW 2170, Australia; 2School of Psychiatry, Faculty of Medicine, University of New South Wales, Sydney, NSW 2052, Australia; 3Sydney Local Health District, Sydney Institute Women, Children and Their Families, Camperdown, NSW 2050, Australia; 4New South Wales Ministry of Health, St Leonards, NSW 2065, Australia; 5Indigenous Health, School of Medicine, Campbelltown Campus, Western Sydney University, Sydney, NSW 2560, Australia; 6South Western Sydney Local Health District, Sydney, NSW 2170, Australia

**Keywords:** developmental screening, screening tools, First Nations populations

## Abstract

Developmental surveillance and screening is recommended for all children under five years of age, especially for those from at-risk populations such as First Nations children. No review to date has, however, evaluated the use of developmental screening tools with First Nations children. This review aimed to examine and synthesise the literature on developmental screening tools developed for, or used with, First Nations populations children aged five years or younger. A PRISMA-compliant systematic review was performed in the PsychInfo, PubMed, and Embase databases. Additional searches were also undertaken. In total 444 articles were identified and 13 were included in the final review. Findings indicated that several developmental screening tools have been administered with First Nations children. Most tools, however, have only been evaluated in one study. Results also found that no studies evaluated actions taken following positive screening results. More research evaluating the accuracy, acceptability, and feasibility of using developmental screeners with First Nations children is required before widespread implementation of developmental screening in clinical settings with First Nations children is recommended.

## 1. Introduction

Developmental difficulties, such as intellectual disorders, communication disorders, specific learning disorders, and autism spectrum disorder, have been linked to a range of negative outcomes across the lifespan [1,2]. Early detection and treatment of developmental difficulties have been associated with improved outcomes [3,4] and, as such, have become a primary goal for many governments [5,6].

Developmental surveillance, together with the use of developmental screening tools, has been shown to increase early identification of developmental difficulties [7,8]. Policy bodies have therefore recommended universal developmental surveillance and screening for all children under 5 years of age, with a focus on children from at risk populations as a means of improving health inequalities [6]. Many, First Nations children are considered developmentally vulnerable [9,10,11] which is at least partly due to the continued health and social disadvantages experienced by First Nations populations relative to non-First Nations populations [12]. Thus, screening for developmental concerns in First Nations populations is crucial.

Although First Nations populations are considered priority populations and the use of developmental screening tools has been recommended as part of the developmental surveillance processes, no review, to our knowledge, has been undertaken to evaluate the use of developmental screening tools with First Nations populations. It would be apposite to do so given that the use of inappropriate tools could lead to over- or under-recognition of developmental difficulties [13], which may increase the likelihood of poor lifelong outcomes (e.g., under-recognition may lead to a delay in accessing treatment services while over recognition may lead to accessing treatment services that are not necessary).

## 2. Aims

To better understand the literature on using developmental screening tools with First Nations populations, this review aimed to answer the following research questions:What developmental screening tools have been administered with, or tailored for use with, First Nations children?What is the accuracy of using the identified developmental screening tool with First Nations children?What is the knowledge, acceptability, and feasibility of using the identified screening tool with First Nations children?What actions have been taken following positive developmental screening?

## 3. Method

Prior to the commencement of this review, a study protocol was developed and registered with the University of York Centre for Reviews and Dissemination (PROSPERO; registration number: CRD42022337320). The study protocol was focused on universal developmental screeners for children under the age of 5 years; however, given the literature identified on First Nations populations, the authors thought it best to divide the results into two separate reviews—a review on universal developmental screeners and a review on screeners for First Nations children.

A narrative systematic review approach was used to examine and synthesise findings. A narrative systematic review was chosen following preliminary searches, which identified that studies and reported outcomes could not be analysed through a meta-analysis. The process included four steps: (1) a systematic search of available literature; (2) a multi-step screening process guided by the inclusion and exclusion criteria; (3) an assessment of the methodological quality of studies; and (4) a qualitative synthesis of selected studies presented in the results section.

### 3.1. Search Strategy

A systematic search of published literature available from inception to May 2022 was conducted. Four search strategies were implemented to identify relevant research studies. First, interdisciplinary research databases PsychInfo, Embase, and PubMed were searched concurrently for any articles containing the following terms: “child” OR “infant*” OR “baby” OR “preschool” AND “milestone*” OR “surveillance” OR “screening tool*” OR “screening measure*” OR “screening assessment*” AND “first nations” OR “indigenous” OR “Aboriginal” OR “native” OR “Torres Strait Islander*”. All searches were limited to entries conducted on “human” subjects and published or submitted for publication in an “English Language”, “peer-reviewed” journal from inception to May 2022. Figure 1 presents the detailed search terms next to the main search terms of “First Nations”, “Child”, and “Screening Tool”. As per the Preferred Reporting Items for Systematic Reviews and Meta-Analysis (PRISMA) guidelines [14], Appendix A provides the search strategy for each database. Second, the reference lists of articles selected for this review were searched manually. Third, internet searches for grey literature were conducted alongside focused searches on key websites including screening tool developer websites. Fourth, cited reference searches were conducted using Google Scholar and PubMed databases to search for any publications that had cited any of the included studies.

#### 3.1.1. Inclusion and Exclusion Criteria

Articles were included if: (1) they evaluated a universal developmental screening tool or a screening tool focusing on at least one developmental domain (e.g., communication); (2) the study sample included First Nations participants aged 0–5 years (if samples include a wide age group they were included if the average child age was below 6 years); and (3) the article was published or submitted for publication in English. The search was not limited by year and included articles from inception to May 2022. Articles that looked at practitioner and parent acceptability of surveillance or screening tools used with First Nations children were also included if they were published in English. Articles were excluded if: (1) they included a screening tool to evaluate an intervention outcome only (i.e., there was no evaluation of the screening tool itself); (2) they used a screening tool to evaluate a specific population only (e.g., children with congenital heart disease); (3) the sample did not include First Nations children; (4) they were not available in English; (5) they were not data-based (e.g., books, theoretical papers, reviews); or (6) they were unpublished dissertations/theses. Furthermore, studies that included First Nations populations but did not segregate data based on ethnicity were excluded as it was not possible to interpret the data on First Nations populations only.

#### 3.1.2. Assessment of Methodological Quality

Two reviewers independently assessed the quality of included studies using the Mixed Methods Appraisal Tool (MMAT; [15]) or the Quality Assessment of Diagnostic Accuracy Studies Tool (QUADAS-2; [16]). The MMAT was used to assess the quality of qualitative research, non-randomised studies, and mixed methods studies while the QUADAS-2 [16] was used to evaluate diagnostic accuracy studies. The MMAT allows for the assessment of the methodological quality of qualitative research, randomised controlled trials, non-randomised studies, quantitative descriptive studies, and mixed methods studies. The risk of bias is determined based on five sources for each study category. For example, for randomised control trials, study quality is based on (1) randomisation; (2) group comparability; (3) complete outcome data; (4) blinding of assessors; and (5) participant intervention adherence. Reviewers are required to assign a “yes”, “no”, or “can’t tell” to each outcome. It is recommended that the “can’t tell” option is used when not enough information is available for the reviewer to assign a “yes” or “no”. The MMAT is, however, not suitable for use with diagnostic accuracy studies. The QUADAS-2 [16] was therefore used to evaluate diagnostic accuracy studies. The risk of bias on the QUADAS-2 is determined based on (1) patient selection; (2) index text; (3) reference standard; and (4) flow and timing. The QUADAS-2 also allows you to assess applicability based on concerns that the study does not match the review questions. Reviewers are asked to rate each risk of bias and applicability concern outcome as “low”, “high”, or “unclear”. Both the MMAT and QUADAS-2 discourage the calculation of an overall quality score, consensus on the quality of studies was therefore reached through discussion (see Table 1 and Table 2 for quality assessments of the included studies).

#### 3.1.3. Data Extraction

The Cochrane Effective Practice and Organization of Care Review Qualitative Evidence Syntheses guidelines were used to inform data extraction [28]. Data items extracted included study design, setting, aims, population, measure being evaluated, who completed the measure, and study outcomes relevant to the review. Data extraction forms were first piloted and amended as necessary. Author PH performed the initial data extraction, which was then reviewed and refined by SC.

#### 3.1.4. Data Synthesis

As indicated above, a narrative approach was used to synthesise results.

## 4. Results

Figure 2 presents an overview of our search strategy. Initial database searches resulted in a total of 442 articles. Title and abstract screening resulted in the exclusion of 431 articles, resulting in 11 articles. Two additional articles were identified through reference list searches, grey literature searches, and cited reference searches, resulting in a total of 13 articles that underwent full-text screening. The thirteen articles met inclusion criteria and were included in the current review. Two reviewers (SC and PH) independently completed title and abstract searches, full-text reviews, and quality assessments. They then compared their results and any disagreements regarding study selection and quality assessment were discussed and resolved. A third reviewer (AMD) was available in case disagreements could not be resolved by the primary reviewers.

### 4.1. Overview of Included Studies

Table 3 presents an overview of studies included in this review. Of the 13 studies, nine were undertaken in Australia [13,17,18,19,20,22,23,25,27], two in the United States of America (USA; [21,24]), one in Canada [26], and one in Peru [11]. Eight studies were cross-sectional, two were qualitative, and three were mixed methods.

### 4.2. Screening Tools Administered with, or Tailored for Use with, First Nations Children

Five studies evaluated the Ages and Stages Questionnaires: Talking About Raising Aboriginal Kids (ASQ-TRAK; [18,19,20,25,27]), three evaluated the Ages and Stages Questionnaires Third Edition (ASQ-3; [11,24,25]), one evaluated the Ages and Stages Questionnaires Second Edition (ASQ-2; [26]), one evaluated the Parents’ Evaluation of Developmental Status [22], one evaluated the Brigance Screens [13], one evaluated the Parents’ Evaluation of Listening and Understanding Measure (PLUM; [23]), one evaluated the Hearing and Talking Scale (HATS; [17]), and one evaluated the Survey of Well-being of Young Children (SWYC; [21]). The majority of identified screening tools were universal developmental screening tools, with the two exceptions being PLUM and HATS which are specific domain screeners.

Further studies, which evaluated the Learn the Signs. Act Early [29,30], and SWYC [31], for example, noted that they had First Nations populations in their participant samples but did not separate data based on ethnicity, and were subsequently excluded from further analysis. Furthermore, grey literature searches revealed that the PEDS, when used as a population level screener, has been administered to Aboriginal and Torres Strait Islander people in Australia [32] and First Nations people in the USA [33].

### 4.3. Accuracy of Identified Developmental Screening Tools

Nine studies assessed the accuracy of a screening tool [11,13,17,18,19,22,23,24,26]. Three examined the accuracy of the ASQ, two the ASQ-TREK, one the Brigance Screens, one the PEDS, one the PLUMS, and one the HATS.

ASQ-3 and ASQ-2. The ASQ assesses communication, gross motor, fine motor, problem solving, and personal-social skills in children aged 0–5 ½ years [34]. Three studies looked at the accuracy of the ASQ when administered with First Nations populations in the USA (Navajo population, measure completed by professionals; [24]), Canada (First Nation Mohawk territory in Eastern Canada, measure completed by parents and teachers; [26], and Peru [Amazonian departments of Loreto and Ucayali, measure administer by professional to parents; 11]. Participants were recruited from a child and family centre [26], Indian Health Services unit [24], and community-based enrolment as designated by mayors of several districts [11]. All three studies made minor modifications to the measure prior to administration. Results suggest that, when modified, the ASQ may be suitable for use with First Nations populations however more research needs to be undertaken before any definitive conclusions can be made.

ASQ-TRAK. The ASQ-TRAK is comprised of seven questions which assesses communication, gross motor, fine motor, problem solving and personal-care domains [20]. The questions were adapted from the ASQ-3 to be more culturally appropriate for screening Australian Aboriginal children aged 2–48 months [20]. Two studies evaluated the accuracy of the ASQ-TRAK [18,19], both studies were undertaken by the same team. The first study, conducted by Simpson, D’Aprano [18], recruited 67 children aged 2-36 months from Alice Springs, Northern Territory. Results showed that scores on the ASQ-TRAK communication, gross motor, fine motor, and problem-solving domains were moderately correlated with the Bayley Scales of Infant and Toddler Development Third Edition (BSITD-III). Similarly, in a subsequent study Simpson, Eadie [19] found the ASQ-TRAK to be moderately positively correlated with the corresponding domains of the BSITD-II or BSITD-III in a sample of 336 Aboriginal children aged 2–48 months recruited from the Northern Territory and South Australia. Their results also showed that the measure had acceptable sensitivity (83%), specificity (83%), and negative predictive value (99%).

Brigance Screens. The Brigance screens are developmental screeners for children aged 0–8 years [35,36,37,38]. D’Aprano, Carapetis [13] were the only team to assess the accuracy of the Brigance screen in a sample of Australian Aboriginal children. The sample was comprised of 124 participants recruited from remote communities in the Northern Territory, Australia. The scale was administered to all participants by a paediatrician. Results showed that all participants scored below the age-specific cut-offs identifying those at risk of developmental disabilities. The authors concluded that the scale was not useful in differentiating children who had developmental delays in remote Australian Aboriginal communities.

PEDS. The PEDS is a developmental screener consisting of 10 questions that assesses for global/cognitive, expressive language and articulation, receptive language, fine and gross motor, behaviour, self-help, socialisation, and academic concerns in children aged 0-8 years [39]. One study was identified that evaluated the PEDS with urban Aboriginal and Torres Strait Islander children in Australia aged younger than 8 years [22]. Results showed that 32%, 28%, and 40% of children screened using the PEDS were at high, moderate, and low/no developmental risk, respectively. The authors indicated that rates of developmental risk were consistent with those observed in higher-risk populations in the USA.

PLUM. The PLUM is a 10-item screener used to assess functional auditory performance in Aboriginal and Torres Strait Islander children aged 6 years or younger living in urban, rural, or remote communities [40]. To date, only one study has evaluated the PLUM [23]. Ching, Hou [23] described the development of the measure and evaluated its psychometric properties with a sample of 438 young children from urban, regional, and remote communities. Their findings showed the measure to have good internal consistency reliability (0.87).

HATS. The HATS screens for communication problems in Aboriginal and Torres Strait Islander children aged 6 years or younger living in urban, rural, or remote communities [40]. The only study to evaluate the HATS was undertaken by Ching, Saetre-Turner [17]. The study described the development of the screening tool and evaluated its psychometric properties using a sample of 68 young children (46 Aboriginal and Torres Strait Islander Children, 22 non-Indigenous children). Results showed that, compared to the ASQ-TRAK and Expressive Vocabulary Tests, the HATS had 80% and 81% accuracy, respectively.

Despite limitations such as only a small number of studies being undertaken assessing the accuracy of developmental screeners for use with First Nations populations and most studies being undertaken in Australia, available evidence indicates that modified versions of the ASQ may be appropriate for assessing developmental difficulties in First Nations populations.

### 4.4. Knowledge, Acceptability, and Feasibility of Universal Developmental Screening Tools

Of the five studies exploring the knowledge, acceptability and/or feasibility of administering developmental screeners, three studies were undertaken in Australia and evaluated the ASQ-TRAK [20,25,27], one study was conducted in Canada and evaluated the ASQ [26], and one was undertaken in the USA and evaluated the SWYC [21].

ASQ. Results from the Dionne, McKinnon [26] study indicated that parents (*n* = 282), recruited from a child and family centre, described the ASQ-2 as fun and easy to use and noted that it helped them become more aware of their child’s abilities. Some participants, however, indicated that the rating procedures were confusing; that visual cues, pictures and symbols could help ease the interpretation of items; and that parents should be made aware that it was normal for a child not to be able to accomplish all items.

ASQ-TRAK. In the three studies that evaluated the ASQ-TRAK, participants were recruited from the Northern Territory, Australia [20,25] and urban, regional and remote communities in South Australia [27], and included parents and caregivers [20,27], Aboriginal health workers [20], child development specialists [20], health practitioners [25], and key community informants [20]. Results from all three studies reported that participants found the measure acceptable for use in their communities. Furthermore, families indicated that they believed the ASQ-TRAK was easy to use [20,27], and health professionals rated the ASQ-TRAK as more acceptable and easier to understand for parents compared to ASQ-3 [25].

SWYC. The SWYC consists of short answer questions relating to child development, child behaviour, and family risk factors (e.g., depression, substance abuse) that are completed by caregivers of children aged 1–65 months [41]. Findings from Whitesell, Sarche [21] study showed the participants (N = 199 staff from Head Start, Home Visiting, and Child Care programs; paediatricians; behavioural health providers; parents of young children; tribal leaders; and other stakeholders in seven diverse communities) appreciated the brevity of the SWYC, that it was free to use and easy to administer and score. Furthermore, they appreciated that it did not require training or lengthy scoring procedures, was comprehensive and provided opportunities for dialogue with parents about child development. Participants also noted that the measure might need to be translated into tribal languages and expressed concerns that the milestones did not reflect native languages or cultural environments. Furthermore, participants indicated that they found other commonly used screening tools, including the ASQ and ASQ: SE, time-consuming and costly. They expressed frustration regarding needing to complete multiple tools across different settings and the uncertainty regarding the appropriateness of developmental tools for First Nations children.

In sum, a strength of this body of research is that parents, various health providers, and community workers (e.g., childcare providers) were included in the study samples. The literature is, however, limited by the small number of studies undertaken and that most available studies were undertaken in Australia and examined the ASQ-TRAK.

### 4.5. Actions Taken following Positive Screening

None of the identified studies commented on whether any follow-up action was taken following positive screening.

## 5. Discussion

To the best of our knowledge this is the first review to identify developmental screening tools which have been developed for, or implemented with, First Nations populations and summarise their acceptability, feasibility, and accuracy in detecting developmental delays in children under the age of 5 years. Our analysis included 13 studies, with most studies undertaken in Australia (*n* = 9) and evaluating the ASQ-TRAK (*n* = 5).

Together the review results indicated that the ASQ-TRAK has the greatest evidence base regarding accuracy, acceptability, and feasibility. Despite the ASQ-TRAK’s promising results, only 5 studies evaluating the measure are available, and all studies were undertaken in Australia with Aboriginal and Torres Strait Islander communities. Given that there is a variety of diverse Aboriginal and Torres Strait Islander communities (i.e., different cultures and different languages spoken) that live in all parts of Australia [42], more research, which encompasses a broad scope of Aboriginal and Torres Strait Islander communities’ needs, are to be undertaken to determine the generalisability of the ASQ-TRAK for use with all Aboriginal and Torres Strait Islander children. Additional research on the ASQ-TRAK may also help increase the uptake of the measure, with recent research showing that despite the promising evidence, most Aboriginal Community Controlled Organisations have not yet accessed the tool or its training [43]. Furthermore, it is important to reiterate that the ASQ-TRAK only has evidence for use with First Nations children in Australia, and the results are therefore not generalisable to First Nations children internationally.

This review’s results also highlight that there is limited literature (*n* = 4 studies) on the use of developmental screening tools in First Nations populations outside of Australia. Of the available studies, the SWYC had favourable qualitative findings [41] and that when modified, studies on the ASQ suggest that it may be appropriate for use with First Nations children e.g., [11,24,25]. No studies, however, reported whether adjustments to cut-off scores were made. Failure to adjust the cut-off scores may increase the likelihood of over- and under-recognition of developmental difficulties in First Nations populations [13]. Thus, further research needs to be undertaken to determine whether adjustments to standardised measures increase the accuracy of detecting developmental problems in First Nations children. In general, the small number of studies assessing the accuracy, acceptability, and feasibility of the use of developmental screeners, especially outside of the Australian context, indicates that more research needs to be undertaken to determine the appropriateness of using available developmental screeners (which is currently standard practice) with First Nations children.

A further finding of this review was that no studies to date indicated whether action was taken following developmental screening or evaluated outcomes of developmental screening in First Nations populations. Information regarding screening outcomes can help determine the benefits and risks associated with developmental screening. Commonly cited benefits associated with early developmental screening often include early diagnosis and access to early intervention [44,45]. Early intervention has been found to support the development of core (e.g., communication) and related (e.g., play) areas of child development [46]. In contrast, a risk frequently associated with screening is that of receiving a false positive screen and the time, effort, and anxiety associated with further testing [47]. A recent qualitative study with 26 culturally diverse, low-income parents whose children had received a false-positive autism screen, however, found that the benefits of screening (e.g., connections with developmental services, more knowledge regarding child development) outweighed the distress associated with a positive screen. Nonetheless, more research evaluating the benefits and risks associated with screening in First Nations children is needed, especially if screening tools developed for Western populations are used as they may not be relevant to First Nations cultures, values, and beliefs [13].

## 6. Clinical and Research Implications

This review indicated that several measures have been used to screen for developmental concerns in First Nations children and that the screeners improve the detection of developmental delays. Furthermore, the studies included in this review suggest that implementing the screeners is feasible and acceptable. While the literature on the accuracy, acceptability, and feasibility of screeners is promising, the majority of identified measures have only been evaluated in one study and with one group of First Nations children, with the exception being the ASQ-TRAK. Furthermore, most studies were cross-sectional in nature. Given the diversity of First Nations populations, more research, implementing more rigorous methodology, on the use of developmental screeners with a variety of First Nations children is needed before widespread implementation of a screener in clinical settings is recommended with First Nations populations. In regard to First Nations children in Australia, the ASQ-TRAK has had promising results, however, further research needs to be undertaken to determine the generalisability of findings.

## 7. Strengths and Limitations

This review has several strengths. First the use of systematic review strategy with broad inclusion criteria as well as four different search strategies increased the likelihood that all relevant literature was identified. Second, two reviewers completed all stages of data screening and assessment as well as the risk of bias assessments, increasing the likelihood that all relevant factors were considered. Third, to our knowledge, this was the first review to evaluate the available evidence on developmental screening with First Nations children. The review also had several limitations. First, searches were restricted to studies written in the English language, reducing the generalisability of findings. Second, the majority of included studies were conducted in high-income, Western countries, this also negatively impacts the generalisability of findings. Only including English language studies conducted in primarily Western countries reduces the generalisability of findings as findings may only be applicable to people who live in what are often referred to as educated, industrialized, rich, and democratic (WEIRD) countries and not to non-Western, less-educated, less-industrialized, poorer, nondemocratic countries [48]. Third, the identified studies lacked methodological diversity, with most studies being cross-sectional.

## 8. Conclusions

Taken together, results of this review found that only a small number of studies have evaluated the use of developmental screening tools with First Nations populations. The ASQ-TRAK was identified as the measure with the greatest evidence base supporting its accuracy, acceptability, and feasibility, however, it was adapted for use with Australian Aboriginal and Torres Strait Islander populations, and therefore, the appropriateness of its use with First Nations populations internationally is unknown. Given the small number of available literature on the use of developmental screeners with First Nations populations, especially outside of the Australian context, more research with diverse First Nations populations, more rigorous methodologies, and assessing the benefits and risks of using developmental screeners with First Nations populations needs to be undertaken, before the widespread implementation in clinical settings of developmental screening is recommended for First Nations populations.

## Figures and Tables

**Figure 1 ijerph-19-15627-f001:**
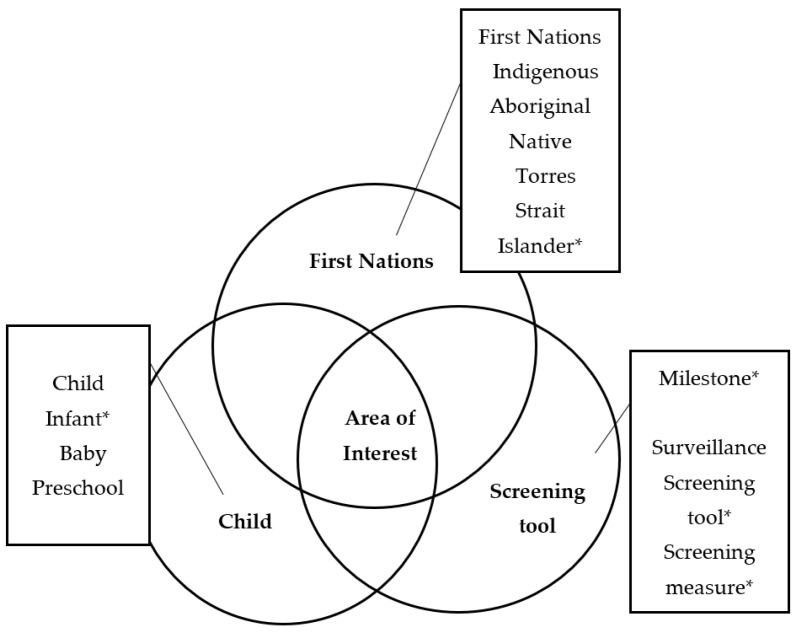
Diagram of search term clusters.

**Figure 2 ijerph-19-15627-f002:**
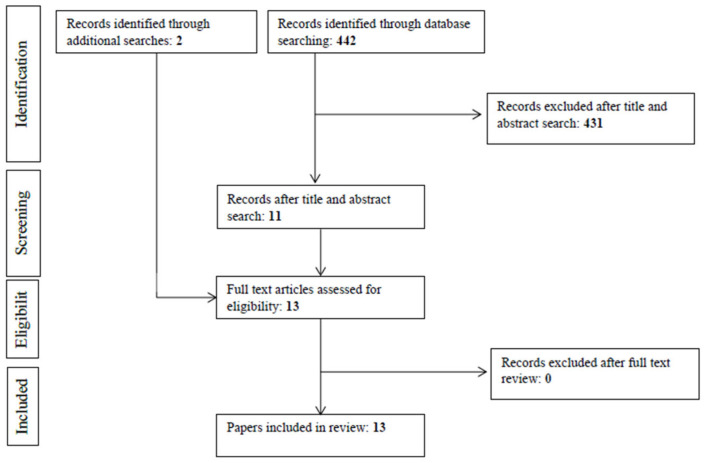
PRISMA flow diagram of included studies.

**Table 1 ijerph-19-15627-t001:** Quality assessment using QUADAS-2.

Citation	Risk of Bias				Applicability Concerns		
	Patient Selection	Index Test	Reference Standard	Flow and Timing	Patient Selection	Index Test	Reference Standard
Ching et al., 2020 [17]	?	?	?	?	L	L	L
Simpson et al., 2016 [18]	L	L	L	L	L	L	L
Simpson et al., 2021 [19]	L	L	L	L	L	L	L

Note. L = Low; H = High; ?—Unclear.

**Table 2 ijerph-19-15627-t002:** Quality assessment using MMAT.

Citation			Criteria		
Qualitative	1.1.	1.2.	1.3.	1.4.	1.5.
D’Aprano et al., 2014 [20]	Y	Y	Y	Y	Y
Whitesell et al., 2015 [21]	Y	Y	Y	?	Y
Quantitative nonrandomised	3.1.	3.2.	3.3.	3.4.	3.5.
Chando et al., 2020 [22]	Y	Y	?	?	Y
Ching, Hou, et al., 2020 [23]	Y	Y	?	Y	Y
D’Aprano et al., 2011 [13]	Y	Y	Y	N	Y
Nozadi et al., 2019 [24]	Y	Y	Y	Y	Y
Westgard & Alnasser, 2017 [11]	N	N	Y	Y	Y
Mixed methods	5.1.	5.2.	5.3.	5.4.	5.5.
D’Aprano et al., 2020 [25]	Y	Y	Y	Y	Y
Dionne et al., 2014 [26]	Y	Y	Y	?	?
Johansen et al., 2020 [27]	Y	Y	Y	Y	N

Note. All studies met MMAT screening questions criteria S1, “Are there clear research questions?”; and S2, “Do the collected data allow to address the research questions?”. Y = Yes; N = No; ? = Can’t Tell. The “can’t tell” option is used when not enough information is available for the reviewer to assign a “yes” or “no”. The question numbers are consistent with those outlined in the MMAT tool version 2018 user guide. Questions: 1.1. Is the qualitative approach appropriate to answer the research question? 1.2. Are the qualitative data collection methods adequate to address the research question? 1.3. Are the findings adequately derived from the data? 1.4. Is the interpretation of results sufficiently substantiated by data? 1.5. Is there coherence between qualitative data sources, collection, analysis and interpretation? 3.1. Are the participants representative of the target population? 3.2. Are measurements appropriate regarding both the outcome and intervention (or exposure)? 3.3. Are there complete outcome data? 3.4. Are the confounders accounted for in the design and analysis? 3.5. During the study period, is the intervention administered (or exposure occurred) as intended? 5.1. Is there an adequate rationale for using a mixed methods design to address the research question? 5.2. Are the different components of the study effectively integrated to answer the research question? 5.3. Are the outputs of the integration of qualitative and quantitative components adequately interpreted? 5.4. Are divergences and inconsistencies between quantitative and qualitative results adequately addressed? 5.5. Do the different components of the study adhere to the quality criteria of each tradition of the methods involved?

**Table 3 ijerph-19-15627-t003:** Overview of studies included in the review.

Citation	Measure	Study Classification	Country	Setting	Person Who Completed Tool	Demographics	Accuracy Evaluated in Study	Knowledge, Feasibility, and/or Acceptability of Measure Evaluated in Study	Follow-up Undertaken in Study
Chando et al., 2020 [22]	PEDS	Cross-sectional	Australia	Primary care	Caregivers	N = 725 children; Age range = 0–8 years; 56% male.	√	−	−
Ching et al., 2020 [17]	HATS	Cross-sectional	Australia	Primary care	Trained staff	N = 68 children. Age range = 0–5 years, 11 months.	√	−	−
Ching, Hou et al., 2020 [23]	PLUM	Cross-sectional	Australia	Childcare	Parents or primary caregivers	N = 438 children; age range 0–100 months; 50% male.	√	−	−
D’Aprano et al., 2011 [13]	Brigance	Cross-sectional	Australia	Community	Trained staff	N = 124 children; age range 3–7 years.	√	−	−
D’Aprano et al., 2014 [20]	ASQ-TRAK	Qualitative	Australia	Community	Community stakeholders	N = 3 key informants, 6 Aboriginal Health Workers, 18 parents.	−	√	−
D’Aprano et al., 2020 [25]	ASQ-TRAK & ASQ-3	Mixed methods	Australia	Primary care	Healthcare practitioners	N = 38 healthcare practitioners, 13 child health nurses, 10 occupational therapists, 4 Aboriginal cultural consultants, 4 physiotherapists, 4 speech therapists, 1 early childhood intervention consultant, 2 not described.	−	√	−
Dionne et al., 2014 [26]	ASQ-2	Mixed methods	Canada	Childcare	Parents or teachers	N = 282 children; age range 9–66 months; 53.9% male.	√	√	−
Johansen et al., 2020 [27]	ASQ-TRAK	Mixed methods	Australia	Primary care	Primary caregivers	N = 99 children. Remote Cohort: *n* = 42; age range = 3.6–48.6 months; 48% male. Regional Cohort: *n* = 14; age range 6.2–48.6 months; 64% male. Urban Cohort: *n* = 43; age range 1.2–49.9 months; 53% male.	−	√	−
Nozadi et al., 2019 [24]	ASQ-3	Cross-sectional	United States of America	Primary care	Trained staff	N = 530; age range 1–13 months; 47.3% male.	√	−	−
Simpson et al., 2016 [18]	ASQ-TRAK	Cross-sectional	Australia	Primary care	Trained staff	N = 67 children; age range 2–36 months; 58% male.	√	−	−
Simpson et al., 2021 [19]	ASQ-TRAK	Cross-sectional	Australia	Community	Parents or primary caregivers	N = 336 children; age range 2-48 months; 52.98% male.	√	−	−
Whitesell et al., 2015 [21]	SWYC	Qualitative	United States of America	Community	Community stakeholders	N = 199 community stakeholders.	−	√	−
Westgard & Alnasser, 2017 [11]	ASQ-3	Cross-sectional	Peru	Community	Trained staff	Children: N = 593; age range 8–39 months. Caretakers: N = 605.	√	−	−

Note. √ = Yes, − = Not described. ASQ-TRAK- Ages & Stages Questionnaires: Talking about Raising Aboriginal Kids; ASQ-2- The Ages & Stages Questionnaires (2nd ed); ASQ-3- The Ages & Stages Questionnaires (3rd ed); HATS- Hearing & Talking Scale; PEDS-Parents’ Evaluation of Developmental Status; PLUM- The Parents’ evaluation of Listening and Understanding Measure; SWYC-Survey of Well-Being in Young Children.

## Data Availability

Data available upon reasonable request.

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
