# Peer review of "Developmental Screening Tools Used with First Nations Populations: A Systematic Review"

_ijerph, 2022, doi:10.3390/ijerph192315627_

Round 1

Reviewer 1 Report

Interesting subject. It can be improved in terms of presentation of data.

In the abstract: “…for those form at risk…”

Tables could be better constructed, avoiding all the text in the head of the columns, which may be put as legenda of the question numbers.

There is a note under table 2 that is not separated from the main text of the paper.

Reviewer 2 Report

This is an interesting, relevant and well-planned systematic review. However, some improvements should be made:

1) At the end of the background information, the research question must appear explicitly; which is specified in the objective and materializes in the forecasts to be obtained in the study (if it were an empirical study we would speak of hypotheses).

2) In the Methodology, a brief section should be opened explaining the procedure followed or phases: 1st search terms and a diagram of search terms would be desirable -see articles by Scott et al for an illustration-; 2nd inclusion and exclusion criteria; 3rd analysis of the quality of the studies; 4th phase qualitative and quantitative results and PRISMA.

3) In the application of the search terms, it would be nice to include a diagram and clarify much better and more clearly the combinations that were made and their results

See as examples of very good Systematic Reviews:

Miller, D. M.; Scott, C. E.; McTigue, E. M. (2018). Writing in the secondary-level disciplines: A systematic review of context, cognition and content. Educational Psychology Review, https://doi.org/10.1007/s10648-016-9393-z   Page, M. J., McKenzie, J. E., Bossuyt, P. M., Boutron, I., Hoffmann, T. C., Mulrow, C. D., Shamseer, L., Tetzlaff, J. M., Akl, E. A., Brennan, S. E., Chou, R., Glanville, J., Grimshaw, J. M., Hróbjartsson, A., Lalu, M. M., Li, T., Loder, E. W., Mayo-Wilson, E., McDonald, S., … Moher, D. (2021). The PRISMA 2020 statement: An updated guideline for reporting systematic reviews. [La declaración PRISMA 2020: una guía actualizada para informar revisiones sistemáticas]. The BMJ, 372(71). https://doi.org/10.1136/bmj.n71

Scott, C. E.; McTigue, E. M.; Miller, D. M. & Washburn, E. K. (2018). The what, when, and how of preservice teachers and literacy across the disciplines: A systematic literature review of nearly 50 years of research. Teaching and Teacher Education 73, 1-13. https://doi.org/10.1016/j.tate.2018.03.010

4) In the explanation of the application of the quality criteria, it should be made explicit if it was applied only to the 13 or if it served as a sieve to include only the 13 that met a minimum quality according to the results of the application of the criteria quality. What was done should be much better clarified.

5) In the presentation of results, a much greater articulation is lacking. For example, around three or four key focuses that advance in the answer to the research question that guides the study, and that approaches the fulfillment of the objective and that accounts for the forecasts made. (It is recommended to consult the articles by Scott et al, to see how the results are articulated in three main focuses or arguments around which all the reviewed studies are analyzed)

6) Quantitative data must also be provided, for example, if applicable, effect sizes, which allow a comparison of the efficacy and efficiency of the studies, for example in some figure

7) The tables should be improved a lot, so that they are more friendly, understandable and balanced (in some columns there is a lot of information and in others almost nothing. They should review their presentation so that the tables are informative at a glance, even providing a figure that reflects what was most accurately found

8) In discussion and conclusions, the research question must be answered, indicate if the objective is achieved and to what extent; indicate whether the forecasts are met; interpret what was obtained in the light of the latest research in the field (2022, 2021); explain much better and in more depth the limitations of the review and its solution paths; theoretical implications should be made explicit; applications for educational and social practice, for example. And their added value or conclusions must be made explicit.

9) The doi must be included in all references or url that allows the direct location of the study

10) The title, abstract and the entire article should be reviewed so that there is more consistency between what is done (answering the research question) and what is provided (effective and limited results).

11) All the changes in the new version that must be uploaded to the application and must be marked in color (not with change control), to facilitate their verification; In addition, what has been done must be explained one by one in the forum to the reviewer.

Authors are encouraged to make all requested changes in order to recommend their publication

Reviewer 3 Report

The authors reviewed the developmental screening tools used with First Nations children. Overall, this review is well written. However, I have some concerns:

1) I would like to know why common databases such as Web of Science and Scopus were not used in searching the literature.

2) With 9 out of 13 studies undertaken in Australia, would it make more sense to only focus on the Australian population?

3) The three aims listed focus on identifying the developmental screening tools and their properties. I would say the contribution of this review is very much limited. What else can the authors offer through this review?

4) 431 pieces of literature were excluded. May I know the reason? (e.g., the percentage for each exclusion criterion).

Round 2

Reviewer 2 Report

The paper is ready for publishing